# The Phonological Mapping Negativity (PMN) as a language-specific component: Exploring responses to linguistic vs musical mismatch

**Jen Lewendon**[1]*, **James Britton**[2], **Stephen Politzer-Ahles**[3]

1 Department of Psychology, New York University Abu Dhabi, Abu Dhabi, United Arab Emirates,
2 Department of Chinese and Bilingual Studies, Hong Kong Polytechnic University, Hong Hum, Hong Kong,
3 Department of Linguistics, The University of Kansas, Lawrence, Kansas, United States of America

* jal9590@nyu.edu

**Data Availability Statement:** All data and analysis scripts are available from the Open Science Framework repository https://osf.io/ec32f/?view_only=6efbcb120e16443381a3a2e9092d3fa8.

## Abstract

The Phonological Mismatch Negativity (PMN) is an ERP component said to index the processing of phonological information, and is known to increase in amplitude when phonological expectations are violated. For example, in a context that generates expectation of a certain phoneme, the PMN will become relatively more negative if the phoneme is switched for an alternative. The response is comparable to other temporally-proximate components, insofar as it indicates a neurological response to unexpected auditory input, but remains considered distinct by the field on the basis of its proposed specific sensitivity to phonology. Despite this, reports of the PMN overlap notably, both in temporal and topographic distribution, with the Mismatch Negativity (MMN) and the N400, and limited research to date has been conducted to establish whether these extant distinctions withstand testing. In the present study, we investigate the PMN's sensitivity to non-linguistic mismatches so as to test the response's specific language sensitivity. Participants heard primes—three-syllable words—played simultaneously to three-note tunes, with the instructions to attend exclusively to either the linguistic or musical content. They were then tasked with removing the first syllable (phoneme manipulation) *or* note (music manipulation) to form the target. Targets either matched or mismatched primes, thus achieving physically identical note or phoneme mismatches. Results show that a PMN was not elicited during the musical mismatch condition, a finding which supports suggestions that the PMN may be a language-specific response. However, our results also indicate that further research is necessary to determine the relationship between the PMN and N400. Though our paper probes a previously unstudied dimension of the PMN, questions still remain surrounding whether the PMN, although seemingly language-specific, is truly a *phonology-specific* component.

## Introduction

Speech processing is a complex yet seemingly effortless process, involving the mapping of highly variable acoustic signals to phonemes, syllables, words, and sentences [1]. For any

**Funding:** This research was supported by two internal grants awarded to SPA and JL (P0038898 & P0034791) from the Hong Kong Polytechnic University (https://www.polyu.edu.hk/en/) The funder had no role in study design, data collection and analysis, decision to publish, or preparation of the manuscript.

**Competing interests:** The authors have declared that no competing interests exist.

model of auditory language comprehension to fully account for this process, it must not only be able to define the various subsystems involved in speech processing, but additionally offer evidence-driven specifications of the temporal parameters surrounding the activation and interaction of these subsystems [2]. Whilst through behavioural measures, for example capturing reaction times, considerable advances in our understanding of the mechanisms underpinning language comprehension have been achieved, such measures reflect not only the variability in targeted cognitive processes, but a myriad of additional mechanisms underlying the response process [3]. Because of this, effects revealed by behavioural measures are often very difficult to attribute to a specific cognitive process [4–6]. Capturing the temporal parameters of the subsystems underpinning speech processing thus necessitates the use of a methodology with exceptionally fine-grained temporal resolution–enter the event-related potential (ERP) technique.

Event-related potentials are 'neural responses associated with specific sensory, cognitive, and motor effects' [7, p4] embedded within electroencephalography (EEG) recordings–which themselves are a relatively coarse measurement of brain activity constituting a multitude of neural sources of activity [7]. The temporal resolution of ERPs allows for investigation into neurocognitive mechanisms and brain activity that unfold from one millisecond to the next. Thus, their wide application to questions of language processing, perception, and production is intuitive, and has enabled attempts to map in fine detail the time-course of ERP component processes within the first second of visual language exposure [8]. Although such accounts outline proposed interactions between the visual and auditory language systems, our understanding of the electrophysiological timeline of auditory language processing lags behind that of its visual counterpart, perhaps in part due to the relative difficulty of capturing consistent responses to highly variable speech input. Nonetheless, decades of research has established a series of reliable responses that occur to auditory language input, including the Mismatch negativity (MMN), Phonological Mapping negativity (PMN), P300, and N400, amongst others. Of these components, the PMN is perhaps the least well understood, despite its seemingly straight-forward characterisation as an ERP response that indexes phonological processing at the pre-lexical level. The PMN typically increases in amplitude in response to an unexpected phoneme, with listener expectations most often generated through task demands or context (e.g., cloze-probability sentences), and is *purported* to be distinct from surrounding language-related negativities including the MMN and N400 on the basis of its functional characteristics. The component is characterised as maximal 300 ms post stimulus presentation, although research documents activity attributed to a PMN response starting as early as 150 ms post stimulus onset [9–11], or as late as 425 ms [12]. In addition to latency, the topography of the PMN effect is also notably varied, and despite most consistently being interpreted as a frontal/ fronto-central component [10, 13–15], consideration of the available literature highlights an effect that most frequently spans frontal, central and parietal sites equally [16]. Crucial to our understanding of the PMN, however, is its purported specific sensitivity to phonological processing–a key characteristic that the present paper serves to test.

ERP components are typically defined through a combination of their topography, latency and functional sensitivity [6]. Notable overlaps in the timing, topography, and functional sensitivity of certain PMN reports with that of surrounding components- particularly the MMN and the N400 –both of which respond on some level to phonological expectancy [14, 17–22]– prompt the question of whether the PMN is indeed an independent, dissociable index of phonological processing. Indeed, a vast amount of early research on the PMN effect was dedicated to disassociating the PMN from the ERP response most commonly associated with speech processing—the N400 [3, 15, 23, 24]. Sensitive to a variety of linguistic and non-linguistic manipulations across a range of input modalities, the N400 typically represents stimulus-related brain

activity 200–600 ms post-stimulus-onset. The response reliably indexes ease of information processing, becoming smaller in amplitude (typically across central or centroparietal scalp regions) when factors render input easier to process [24]. As one of the most extensively utilised and documented ERP responses within language research, the N400 has been found to reliably index semantic proximity, cloze probability, and phonological processing, amongst a host of other processing mechanisms. Despite considerable efforts to distinguish the PMN from the N400, a series of methodological limitations in this research render our understanding of the PMN as either an independent response, or a mischaracterisation of early N400 effects uncertain (see [25, 26] for a review).

Two main paradigms have traditionally been employed in attempts to disentangle the PMN from the N400. The first of these typically uses sentence stimuli with highly predictable final words, for example the phrase 'The piano was out of. . .' which naturally lends the listener to predict the final word 'tune', whilst manipulating the semantic and/or phonological congruency of a critical target word [9, 23, 27, 28]. Utilising such a design, Connolly & Phillips published, in 1994, one of the first studies to propose a distinction between the PMN and N400 on the basis of lexicality. In their study, the authors manipulated the final target word so as to create 4 conditions in which the final target word was 1) fully expected (e.g., "The piano was out of *tune*"), (2) semantically incongruent but initially phonologically congruent (e.g., "The gambler had a streak of bad *luggage* (luck)"), (3) semantically congruent but phonologically unexpected (e.g., "Don caught the ball with his *glove* (hand)"), or (4) semantically and phonologically unexpected (e.g., "The dog chased our cat up the *queen* (tree)"). Measuring the PMN as the most negative point 150–350 ms post-stimulus onset, and the N400 between 350–600 ms, the authors reported increased PMN amplitude for phoneme mismatch (conditions 3 & 4), and an N400 increase for semantic mismatch (conditions 2 & 4). However, it is crucial to consider that, in the given paradigm, the supposedly independent manipulation of semantic incongruency (condition 2) cannot be achieved without simultaneous phonological incongruence. Whilst an overlap in the initial phoneme(s) may delay any such response, the unexpected phoneme nonetheless still occurs at the stage at which the target word deviates from the expected input (i.e., lu**ggage**). An earlier negativity (reported by Connolly & Phillips as a separate PMN response) therefore seems naturally attributable to the earlier onset of the incongruency, whilst the delayed response (attributed to the supposedly separate N400) might just be explained by the later deviation from the anticipated target word. Questions regarding the reliability of PMN responses elicited via these means are further exacerbated given the results of Poulton and Nieuwland [29], who failed to reproduce the results of Connolly and Phillips (1994) in a higher-powered replication with more than double the number of participants of the original study. Instead of producing evidence for a PMN effect as a marker of phonological prediction, Poulton and Nieuwland interpreted their results as demonstrating early auditory N400 effects. For a critique of further experiments using related paradigms, alongside an alternative, yet equally plausible explanation for the presence of a supposedly distinct PMN response pertaining to alpha wave contamination, see Nieuwland [25].

The second method typically employed to elicit separable PMN and N400 effects involves phoneme deletion. For example, Newman et al., 2003 [30] (and see also [14] for later use of a similar paradigm), instructed participants to mentally delete the first consonant of an auditorily presented CCVC word (e.g, clap, /k/). After a brief pause, participants heard either a correct target word, or incorrect target words that formed one of three conditions (1) a wrong consonant condition (e.g., cap); (2) a consonant cluster deletion (e.g., ap); or (3) an irrelevant word (e.g., nose). The authors predicted that if the PMN was a marker for phonological relatedness, a graded response would occur, such that the largest negativity would occur in the incorrect word condition, followed by the wrong consonant and consonant cluster conditions.

Whilst the PMN was found to be significantly reduced in the correct deletion condition, the amplitude increase across wrong consonant, consonant cluster deletion, and irrelevant word conditions was indistinguishable. Importantly however, the authors acknowledged that likely P300 contamination in the correct condition (probably resulting from the relative infrequency of matching vs mismatching targets) potentially reduced the PMN amplitude in the match condition. In a 2020 review, Lewendon et al. note that such contamination theoretically leaves the study with three reliable conditions remaining, for which no significant PMN amplitude difference was reported. Furthermore, the equivalent PMN response for both word (e.g., cap/ nose) and non-word (e.g., ap) targets prompted the authors to conclude that the study presented no lexical/semantic processing demands, and thus that elicitation of a PMN response in a context that did not necessitate semantic processing distinguished it further from the N400. However, Lewendon et al. highlight that the notable absence of N400 reporting renders it difficult to determine whether, in the given paradigm, the N400 would have indexed lexicality distinctions. Without the reliable measure of semantic processing that N400 would offer, the absence of lexicality effect on the PMN amplitude cannot necessarily be deemed meaningful. As Nieuwland notes in his 2019 review [25], the most persuasive evidence for the independence of the PMN effect would be a distinct negative peak, dissipating prior to the onset of the N400, that presented with consistent characteristics across conditions and experiments. Such a finding is not what studies currently show, and may be unlikely given the temporal overlap of the PMN and N400 responses (p. 390).

Questions regarding the PMNs independence from a second temporally proximate response, the Mismatch negativity (MMN) have perhaps only arisen more recently. The MMN is a component functionally tied to detection of auditory prediction or expectation, and was first reported in 1978 by Näätänen, Gaillard, and Mäntysalo [31]. The effect is typically characterised as a frontocentral component peaking between 150–250 ms post stimulus onset [32, 33], thus overlapping entirely with the topography of classic frontocentral PMN reports, whilst additionally sharing substantial temporal overlap with early PMN research. The MMN is most commonly elicited via oddball paradigms in which rules–established via a series of stimuli ('standards')–are violated through the presence of infrequent items ('deviants') [34]. For example, in a string of 'standard' phonemes such as "/p/ /p/ /p/ /p/", the presence of the 'deviant' "/t/" would elicit an MMN response. Given that the method used to elicit MMN responses is, on the surface, clearly different from that used to elicit PMN effects, there has been little inquiry into whether the two responses are truly independent components. The apparent differences between the elicitation of the MMN and the PMN are actually, however, rather superficial. That is, whilst the oddball paradigm differs substantially in its means of generating listener predictions via habituation to that of PMN designs, which typically elicit predictions through context or task instructions, both means ultimately produce a context in which listener-generated predictions about upcoming auditory forms are either violated or met [35]. As such, it is nonetheless important to consider that both ERP components result from the same underlying event—a mismatch between listener-generated predictions and actual auditory input. Albeit minimally explored, the shared functional sensitivity alongside temporal and topographical overlap of these two components invite questions as to whether frontocentral PMN effects might actually reflect a response with similar underlying mechanisms to the MMN.

As noted, the use of ERP methodology to investigate the subsystems underpinning speech comprehension (and their temporal parameters) continues to offer a notable means to vastly extend our understanding of such processes. However, given their use as dependent measures, ascertaining the functional sensitivity of ERP components is vitally important to ensure the validity and reliability of a range of academic and clinical research output. Despite this

pervasive lack of clarity regarding the independence of the PMN from the temporally proximate N400 or MMN responses, what remains integral to the effect is its supposed specific sensitivity to phonological manipulations. However, this selective sensitivity to phonological violations has not, to-date been explicitly tested. Take, for example, the N400. Given that N400 activity is consistently reported for unexpected linguistic events (e.g., words, phonemes), an early question regarding the component's functional sensitivity was whether it might respond within other structured domains, such as music [24]. Albeit an ongoing debate, neural overlaps in the processing of speech and music [36–38], alongside correlations between musical experience and language processing [39, 40] imply underpinning mechanisms that may be inextricably intertwined. Notably however, considerable research has consistently demonstrated that the N400 appears to be insensitive to musical prediction violations or incongruencies. For example, in one of the earlier studies to explore the elicitation of the N400 through non-linguistic prediction, Besson and Macar [41] presented participants with familiar scales (note progressions) and melodies that featured notes which were either correct (congruent with the known melodic pattern/scale) or incorrect (diverging from the expected melody/scale). They reported no N400 response for incongruent notes whilst, in the same experiment, reproducing the standard N400 effect for semantic unexpectedness in linguistic stimuli. Further research has replicated such null effects, typically reporting a P3 response to note violations in place of an N400 effect [42–44] (See Calma-Roddin and Drury (2020) for recent evidence of N400 responses for note violations in familiar melodies, in the absence of such a response for unfamiliar melodies–an effect the authors note to suggest that music violations alone do not influence the N400 effect.).

Concluding that an ERP component is selectively sensitive to linguistic manipulations necessitates its testing in contexts that elicit both linguistic and non-linguistic input. As noted above, the PMN is typically elicited via two main paradigms–a sentence/target word manipulation (herein referred to as 'sentence paradigms'), and a phoneme deletion design ('phoneme paradigms). Although framed as an N400 study, the research of Besson and Macar (1987), alongside others, strongly suggest that a PMN for musical stimuli would not occur in a sentence paradigm. That is, Besson and Macar's research designs entailed target stimuli of notes incongruent with a tune/scale, or phonemes (via that means of unexpected words) incongruent with the sentence context. Whilst the linguistic violation reportedly produced an N400, the musical violations showed no N400 modulation, instead consistently producing a P300 effect. Whilst their design unfortunately didn't include conditions to distinguish between semantic vs. phonological mismatches, the absence of any negative modulations within the 250–350, or later 350–650 ms windows suggest that the no PMN effect occurred for such musical violations. Crucially however, due to the aforementioned issues with interactions between phonological and semantic expectancy in sentence paradigms, the presence of a PMN effect for linguistic manipulations in Besson and Macar's study cannot be conclusively determined.

In the present study we therefore test this proposed phonological specificity in a phoneme paradigm by comparing ERP responses to note or phoneme violations generated by physically identical stimuli. Such an investigation offers insight in two respects. Firstly, in contrast to the surrounding language-related negativities, no investigations have been conducted to ascertain whether the supposed phonological-specificity of the PMN is indeed accurate. The present study will thus serve as the first direct comparison of PMN response to phonological and non-linguistic violations. Such an investigation is crucial to our understanding of the component's use in language research, and its present characterisation as a language-specific response. Secondly, whilst the presence or absence of a PMN effect for non-linguistic incongruence cannot conclusively serve to prove or disprove its independence from the N400 or MMN, it may offer insight into the extent to which the mechanisms that give rise to the component differ from

those of the MMN or N400. As outlined above, the N400 has been widely reported as insensitive to note or pitch prediction incongruence, with studies typically reporting P300 responses in place of N400 modulation. In contrast, the MMN has been reliably demonstrated to occur for musical key progressions, pitch, and timbre violations within an oddball paradigm [45–52]. As such, the elicitation of a PMN response for pitch violations that proves similar (in topography, polarity, and latency), to that of a phonological-mismatch responses, might imply underlying cognitive mechanisms that share similarities with the MMN. In contrast, if pitch manipulations do not produce a PMN effect, instead perhaps eliciting a P3-esque response, one might assume that mechanisms that generate a PMN effect might overlap to a greater extent with those of the N400, given their shared insensitivity to musical incongruencies.

Using identical stimuli across conditions, we thus sought to determine whether the PMN is speech-specific, by comparing responses to word and pitch prediction violation. For the linguistic manipulation, participants were asked to delete the first phoneme from the three-phoneme word (e.g., 'piranha', /pɪˈrɑːnə/), resulting in a predicted nonword target (e.g., 'ranha', /rɑːnə/). Target items either matched (e.g., 'ranha', / rɑːnə/) or mismatched (e.g., 'nana', /nɑːnə/). Simultaneously to each syllable, a note was played (e.g., the sequence C (261.6 Hz), B (246.9 Hz), A (220 Hz). For the note manipulation, participants were directed to delete the first note from the three-note sequence prior to being presented with a target that either matched (B, A) or mismatched (E (164.81 Hz), A) the note pattern. Directing participant attention to either the notes or linguistic stimuli, we therefore manipulated either note or phoneme match/mismatch, whilst retaining physically identical target stimuli across conditions (see Fig 1). Stimuli were additionally counterbalanced such that each target featured (based on prime pairing) as a phoneme match, phoneme mismatch, note match and note mismatch. We predicted that if the PMN is insensitive to non-linguistic musical mismatches, prediction violations for phonemes should influence the component, whilst note violations should not. Such an effect should manifest as a significant difference between the phoneme match and phoneme mismatch conditions, such that mean PMN amplitude for the phoneme mismatch condition should be significantly more negative as compared to the match condition, whilst no significant difference should be present for note mismatch vs note match conditions. Alternatively, if the PMN is perhaps more domain-general, we would expect a significant difference for both phoneme and note mismatch conditions as compared to the match condition, manifesting as significantly more negative mean amplitudes for phoneme and note mismatch conditions vs the match condition.

## Methods

### Participants

Fifty-four native English speakers took part in the experiment. Four participants were excluded due to data noise ($< 75\%$ of trials retained subsequent to artefact rejection, irrespective of response accuracy). Fifty participants were retained in the final analysis (31 female and 18 male, 1 non-binary, mean age = 39.3; $SD$ = 9.4,; range 20–55) with normal or corrected-to-normal vision, no learning disabilities, and self-reported normal hearing. All participants gave written informed consent before taking part in the experiment. Experiment methods were approved by the Human Subjects Ethics office of the Hong Kong Polytechnic University (application no. HSEARS20210812003). All spoke English from birth at home as their native language. Two participants were raised in bilingual households but all considered their proficiency in English to be equal or superior to that of their other language. Forty-six participants were right-handed, whilst three were left-handed and one ambidextrous (Exclusion/inclusion of these participants did not change the pattern of results.). Participants had on average 3.2

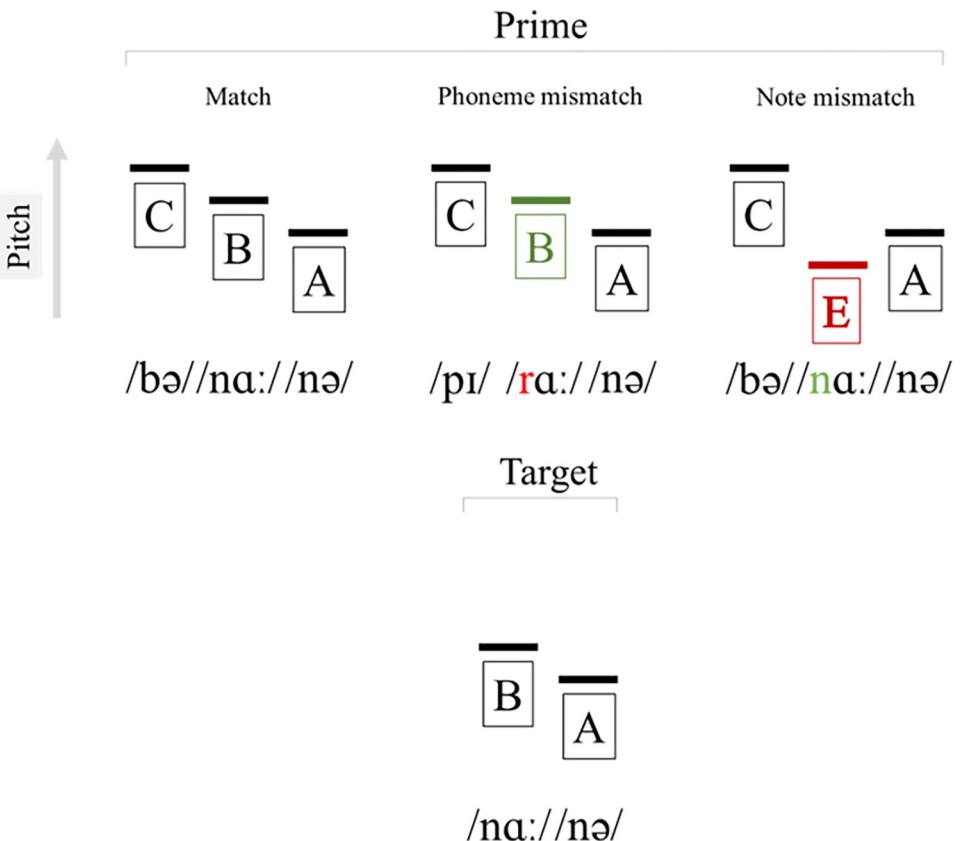

**Fig 1. Experimental manipulations.** +/- correspond to match or mismatch with the target. Prime-target pairings form the following conditions: match (+phoneme +note), phoneme mismatch (-phoneme +note) and note mismatch (+phoneme -note).

years musical experience (*SD* = 4.8), with most participants' experience limited to childhood lessons, and 1 participant playing an instrument casually as an adult. For full participant demographics, see S1 File.

## Stimuli

Stimuli creation followed that of Lewendon, Britton & Politzer-Ahles [35]. Forty trisyllabic words with 2nd syllable stress (e.g., ba*na*na) formed the primes, whilst targets were corresponding prime items without the first syllable (e.g., nana). Prime words had a mean Subtlex Zipf frequency of 3.56 (*SD* = 0.84), and correct/incorrect targets were formed by selecting prime words with identical final syllables, but which differed in their initial two syllables (e.g., bana**na**, pira**nha**). Doing so enabled the creation of 40 target nonwords that formed both match (banana–nana; piranha—ranha) and mismatch (banana–ranha; piranha—nana) conditions through their pairing with one of two primes. Played simultaneously to the word primes were tunes consisting of three- notes that followed fairly simple melodic patterns. Each note coincided with the onset of the first phoneme of each syllable. Example stimuli are given in Table 1.

Using the aforementioned combinations, we created four conditions (i) correct note & correct phoneme (e.g., ambitious–*bitious*; A4-B4-C5 – *B4-C5*); (ii) correct note & incorrect phoneme (e.g., ambitious–*licious*; A4-B4-C5 – *B4-C5*); (iii) incorrect note & correct phoneme (e.g,

**Table 1. Example stimuli.** Note that to ensure full rotation across conditions, both correct and incorrect target words were paired with correct and incorrect target notes.

| Prime word | English IPA | Target word (correct) | Target word (incorrect) | Prime tune | Target tune (correct) | Target tune (incorrect) |
|---|---|---|---|---|---|---|
| infecting | /ɪnˈfek.tɪ/ | Fecting | jecting | G#3-**C#4-F4** | C#4-F4 | G#4-F4 |
| injecting | /ɪnˈdʒek.tɪ/ | Jecting | fecting | G4-**G#4-F4** | G#4-F4 | C#4-F4 |

ambitious–*bitious*; A4-B4-C5 – *E5-C5*); (iv) incorrect note & incorrect phoneme (e.g., ambitious–*litious*; A4-B4-C5 – *E5-C5*). The nature of the design meant that for counterbalancing purposes, participants experienced two additional conditions that were not of relevance to the present study (correct note & incorrect phoneme in the blocks in which participants focussed on the tune, and incorrect note & correct phoneme in the blocks in which participants were directed to attend to the word). Word primes were recorded by a native British female speaker in a sound attenuated room using a cardioid condenser microphone (Audio-technica AT2035, 44100 Hz sampling frequency). Noise reduction was conducted on audio files (freq. 12 dB, sensitivity 6, frequency smoothing bands 3), amplified by 11.292 dB in Audacity® (Version 3.1.3, 1999–2021 Audacity Team) [53]. Subsequently, intensity was averaged at 70 dB (mean power in air: 9.99999357e-06 Watt/$m^2$), and the continuous recording was segmented at zero-crossings to create three individual recordings for each syllable in Praat [54]. Note stimuli were generated in Sibelius® [55] (44100 Hz sampling frequency), amplified (11.292 dB), and segmented such that each note consisted of an audio file 85% of the shortest matching syllable segment duration in any match/mismatch pair. For example, taking the given targets ***fect**ing* and ***ject**ing* from Table 1, if the syllables *fect* and *ject* were 100 ms and 120 ms respectively, both notes C#4 and G#4 would be segmented to 85ms duration. The rationale for reducing note duration was to ensure that the processing of phonological input was not impeded by the notes. Given their constant intensity across the syllable, the presentation of consecutive notes without break was deemed to render the phonological input (with its natural peaks and troughs in intensity) more difficult to distinguish. The mean intensity of each note was matched to the corresponding segment intensity (e.g., *in* & G#3 = 60dB, *fect* & C#4 = 70dB, *ing* & F4 = 80dB), and fade-out was applied to initial and final 50ms of each individual note recording. Tune and word sound files were then combined, and 10 ms ramp up/down was applied to the start and end of all prime and target stimuli.

## Procedure

The experiment consisted of two parts, one in which participants directed their attention to the linguistic stimuli, and a second in which they focussed on the note stimuli (order counterbalanced across participants). Crucial to auditory ERP research, we ensured minimal audio latency jitter by performing the experiment via Presentation® software (Version 18.0, Neurobehavioral Systems, Inc., Berkeley, CA), and used the Presentation mixer Exclusive mode on a Dell Precision T1700, Windows 7 Professional (x64) with a Sound Blaster Z (Creative Technology) sound card, and a Cedrus StimTracker to send trigger events.

Both sections of the experiment started with a brief practice block of 10 items during which participants received auditory feedback about their task accuracy. Following the practice session, each section (phoneme or tune) consisted of 8 experiment blocks of 20 items, with all auditory input played through noise-cancelling foam earbuds. Within each block 10 items formed the present study whilst 10 formed a separate study not reported here (see Table 2).

Participants were asked to mentally delete either the initial syllable/note of a heard prime. They were then instructed to retain the predicted target in their mind, paying particular attention to the (new) initial sound of the item. In any given trial, participants first saw a fixation

**Table 2. Conditions and trials included in the present experiment (note P = phoneme, N = note, +/- indicate match or mismatch).**

| Experiment Section | Condition | Incl. in present study | Condition name |
|---|---|---|---|
| Phoneme-focus | P+N+ | Y | Phoneme match |
| | P+N- | N | NA |
| | P-N+ | Y | Phoneme mismatch |
| | P-N- | N | NA |
| Tune focus | P+N+ | Y | Note match |
| | P+N- | Y | Note mismatch |
| | P-N+ | N | NA |
| | P-N- | N | NA |

cross (200 ms), which stayed on the screen during the presentation of an auditory prime. At the offset of the auditory prime, a variable ISI between 1900 and 2100 ms occurred to enable participant to process the sound deletion and generate predictions of the auditory target. Participants then heard the target (average duration 830 ms, SD = 120 ms), and following its offset indicated via button press whether the trial was a match or mismatch. Button press immediately triggered the start of the subsequent trial. Prime pairs (e.g., infection/injection, and their corresponding tune) and prime repetitions with different target pairings (e.g., infection–fection; infection–jection) were separated by block. Block and stimuli order was randomised. For full stimuli rotations and audio files, see S2 File.

## Data acquisition & pre-processing

Prior to recording, a cap was fitted to secure the EEG electrodes in place on the scalp at specific locations according to the extended international 10–20 system. Electrode impedances were reduced via abrasion to $< 5 \, k\Omega$. Two bipolar facial electrodes were placed lateral to the outer canthi of each eye, whilst electrodes on the inferior and superior areas of the left orbit provided recordings of the horizontal and vertical electrooculograms (EOG). Data were recorded with an online reference located between Cz and CPz. All data pre-processing was conducted in MATLAB, aided by EEGLAB [56], and first involved the application of a high-pass filter offline using 0.1 Hz (half-amp -6dB) IIR Butterworth Zero Phase shift filter, with a 24 db/oct roll-off and DC offset removed. Data were subsequently cleaned via visual inspection to remove pauses and breaks, then offline re-referenced using the average of the left and right mastoids as the new reference. Ocular correction was conducted via Independent Component Analysis (ICA) using the RUNICA algorithm, and an average of 1.9 components removed per participant (range: 1–3). Data were subsequently low-pass filtered at 30 Hz (24 dB/oct) IIR Butterworth Zero Phase shift filter, segmented from −200 ms pre-stimulus to 800 ms post-stimulus (timelocked to the onset of the auditory target), and baseline correction was conducted based on the 200 ms pre-stimulus activity. Automatic artifact rejection was conducted to remove extreme values (±100 μV), resulting in a total of 274 trials removed, and incorrect responses were excluded leaving an average of 38.48/40 (sd = 2.68) trials in the correct phoneme condition, **38.74**/40 (sd = 2.35) trials in the incorrect phoneme condition, 38.54/40 (sd = 3.47) trials in the correct note condition, and **38.76**/40 (sd = 2.25) trials in the incorrect note condition.

## Analyses

Great variation in previous reports of PMN characteristics [14, 30] rendered selection of a priori topography and latency predictions difficult. Instead we took the approach of investigating the data via spatiotemporal cluster-based analyses [57]. This analysis procedure involved

conducting paired t-tests at every channel and every sample from 150 to 550 ms for the contrasts between both phoneme and note mismatch and match conditions, forming clusters of spatiotemporally adjacent datapoints with significant tests (with uncorrected one-tailed alpha < .05 for the 1-tail phoneme mismatch analysis, and corrected one-tailed alpha < .05 for the 2-tail note mismatch analysis), selecting the strongest cluster (based on the sum of t-statistics within the cluster), and evaluating the statistical significance of that cluster via a permutation test.

## Results

Although both note and phoneme match/mismatch were manipulated throughout the experiment, analyses were only run on full matches (i.e., both phonological and tonal properties of the target word were expected, but participant attention was directed to one or the other, to create two identical conditions), or on mismatches that occurred for only one of the two target word features (i.e., phonological mismatches that were heard alongside note matches, and note mismatches heard alongside phonological matches). Restricting the analyses in this sense enabled independent manipulation of note and phoneme expectancy.

### Behavioural results

Overall accuracy was 93.8% in the phoneme match condition, 89.4% in the note match condition, 59.5% in the note mismatch condition, and 75.1% in the phoneme mismatch condition (When excluding low-accuracy participants and/or incorrect trials the same general qualitative pattern of results was obtained but didn't always reach statistical significance (attributed here to lack of power). As such, whilst the task itself was undoubtedly more difficult than anticipated, accuracy did not seem to significantly influence ERP responses to match/mismatch conditions.).

### ERP results

Visual inspection of the ERP results revealed, as expected, more negative ERPs for phoneme mismatches vs phoneme matches across central and parietal sites. In contrast, note mismatched elicited a more positive response as compared to note matches across frontal and central regions. The spatiotemporal cluster-based analysis confirmed these observations, revealing a significant effect of phoneme mismatch, such that ERPs in response to phonologically mismatching were significantly more negative than phonologically matching targets ($p$ = .008). The increase in negativity most pronounced across central, parietal and occipital electrodes (Fig 4A). A second spatiotemporal cluster-based analysis for the note manipulation showed no negative differences between note match and note mismatch conditions ($p$ = 0.219). In contrast a significant positive difference was found between conditions, such that note mismatch elicited a more positive response than note match ($p$ = 0.028) (Fig 4B), which was most pronounced over frontal, frontocentral and central electrode sites. A final exploratory analysis of the phoneme effect revealed no significant positive differences between the phoneme match and mismatch conditions ($p$ = 0.086). Waveforms and raster plots for the respective contrasts can be seen in Figs 2–4.

## Discussion

In the present study, we investigated whether the Phonological Mapping Negativity component could be elicited by non-linguistic stimuli, exploring the supposed language-specificity of the response. We created phoneme and note mismatches using identical target stimuli across

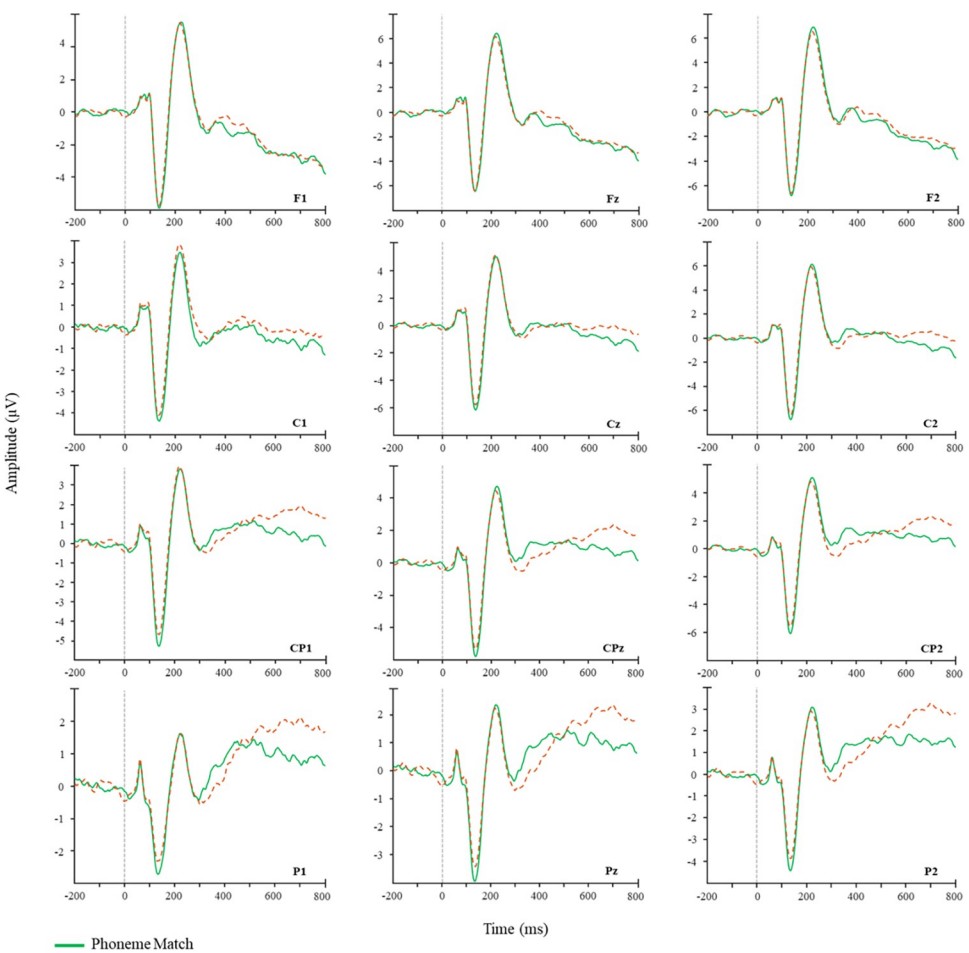

**Fig 2. Waveforms for the phoneme match vs phoneme mismatch conditions.**

conditions to enable a direct comparison of the electrophysiological responses to each mismatch type. Our results revealed the predicted negativity for phoneme mismatches across centroparietal sites, whilst note mismatches elicited a slightly earlier, positive effect across frontocentral sites. The lack of PMN response to note mismatches suggests that the PMN does not respond to prediction mismatches in non-linguistic structured domains, such as music–a finding would suggest that the component may be language-specific (Note that the other major claim about the PMN is that it is insensitive to lexicality. It is important to note here that whilst we interpret our results here as evidence for the PMNs specific sensitivity to language, we refrain from making any claims as to the feature(s) of language the PMN may or may not be selectively sensitive to, as the effect's insensitivity to lexicality was not tested in this study and remains a distinction that we feel requires further investigation.). Such a finding corroborates original interpretations of the response as a language-related negativity [10, 11, 13, 15, 30]. This said, although not supported by any statistical tests or analyses of the present study, visual observation of the topographical plots for the phoneme and note mismatch effects do suggest a certain degree of similarity between the responses. Both effects clearly show a frontal positive and posterior negative pattern. Whilst such an inversion of effects–with components consisting of opposite polarities on one side of the head compared to the other is typical of all ERP responses, the similar dipoles but slightly different orientations of the

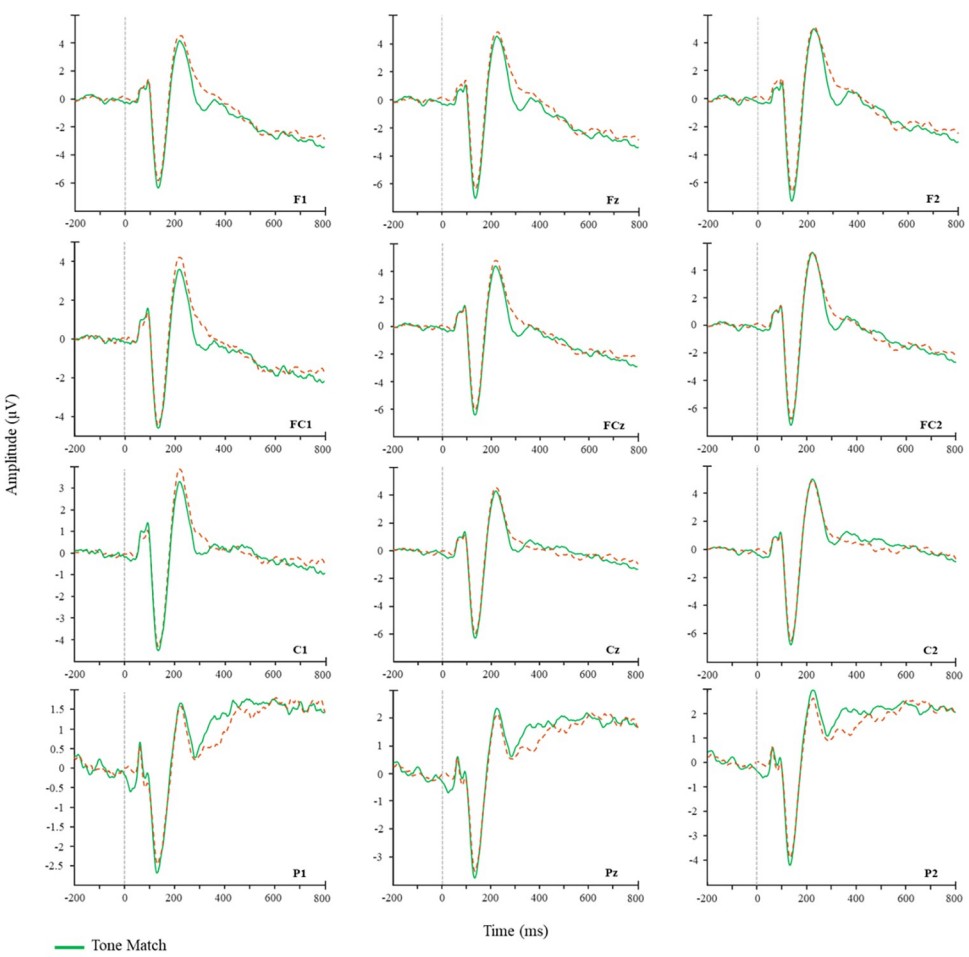

**Fig 3. Waveforms for the note match vs note mismatch conditions.**

topography plots perhaps warrant further investigation. It remains plausible that the negativity occurring to tune mismatch may just have been inadequately captured by the present methodology and, although we limit speculation regarding this visual observation, future work may wish to consider whether the responses reported here are more similar than the present methodology was able to detect.

In any interpretation of our findings, it is important to recall that the PMN remains a contentious component, with unresolved questions regarding its topography, timing, functional sensitivity and, crucially, independence from its surrounding components. Thus, given the outstanding issues surrounding the characterisation of the PMN, we feel that our findings are perhaps most impactful when considered in light of their contribution to ongoing debates regarding the independence of the PMN from its surrounding language-related negativities. In a recent account, Bornkessel-Schlesewsky and Schlesewsky [58] posit that language-related negatives (i.e., MMN, N400, LAN) might represent a family of related components, with shared underlying mechanisms intrinsically related to precision-weighted predictive coding error. The underlying concept that forms the foundation of the Bornkessel-Schlesewsky & Schlesewsky account is that—in an attempt to actively construct explanations for real-world sensory input—the brain continually hypothesises causes for sensory consequences. These hypotheses are compared to actual sensory input, and mismatches between the two give rise to

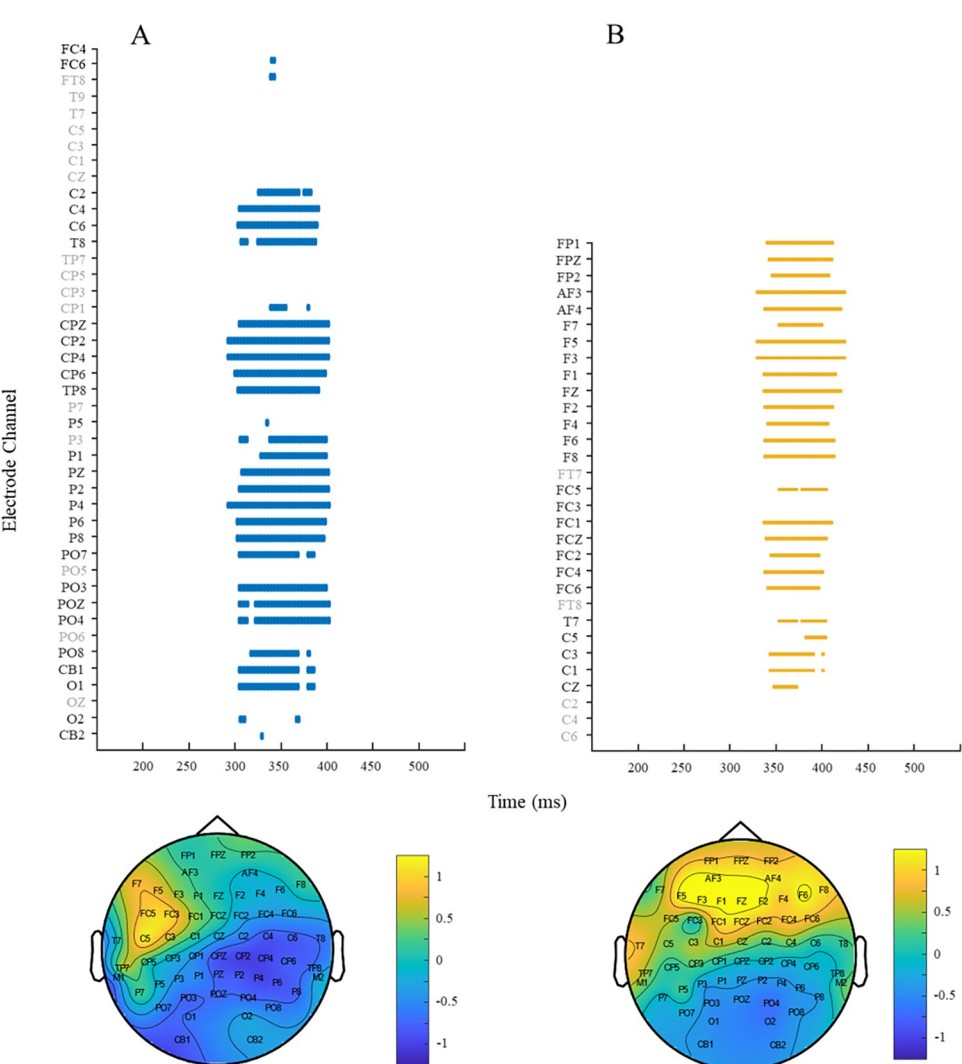

**Fig 4.** Raster plots for cluster test run on (A) phoneme match vs phoneme mismatch, and (B) tone match vs tone mismatch. Note that electrodes labels in black denote electrodes that formed part of the cluster that drove the most pronounces difference between conditions. Topography plots represent the time periods as denoted by the clusters (phoneme: 297–401 ms; and note: 259–348 ms).

a prediction error. These errors prompt updates to the internal model, with the process resulting in a highly efficient coding scheme in which only discrepancies between the internally generated hypotheses and actual input need to be conveyed [58]. The accounts suggests that clear functional links between the MMN and N400 render it possible that the N400 is, in essence, a long-latency MMN, with topographical and latency distinctions originating from the complexity and nature of the predictive coding error as opposed to indicating distinct cognitive processes.

Whilst we do not dispute the plausibility of the Bornkessel-Schlesewsky account (and in fact propose it likely that the PMN falls within this family of precision-weighted predictive coding error responses), we do agree with the authors that further work is necessary to verify the theory. One line of research that may prove insightful is to determine the degree to which the functional sensitivities of these negativities overlap and differ. As reiterated throughout

this paper, the PMN is known to share a number of functional links, alongside topographical and latency characteristics with both the MMN and N400. How then, could we expect the PMN to respond if it is–in fact–either a delayed-latency MMN response or a rapid-onset N400 effect [25, 26, 59]? The MMN is known to be sensitive to a range of linguistic stimuli and, importantly, similarly responsive to non-linguistic stimuli such as musical notes, tones or chords. For example, Tervaniemi et al. [47] played participants continuously looped streams of four 125 ms notes, constituting the middle octave of a given Western musical scale (e.g., . . .E–F–G–A–E–F–G–A. . .). The authors investigated two manipulations, (1) the encoding of the temporal order of note pattern, by introducing infrequent changes embedded within the note stream via reversal of the order two consecutive note (E–G–F–A); and (2) encoding of the pitch of the notes, by replacing frequent note sequences that formed a given Western major chord (e,g, C major. . .C–E–G–C. . .) with infrequent sequences that formed a minor chord (e. g, C minor. . .C–E flat–G–C. . .). Both trained musicians and non-musicians produced an increase MMN response for temporal order and pitch changes, although responses to changes in temporal structure were significantly larger for musicians than non-musicians. Further research has similarly reported MMN responses for musical timbre [47, 48, 52], and pitch [49–51, 60]. Such findings imply that if the PMN and MMN may respond very differently to non-linguistic violations. This distinction between the cognitive mechanisms underpinning PMN and MMN responses is strengthened by a recent paper in which two experiments (including the inattentive data from the present study) demonstrate that the PMN does not appear to be elicited inattentively–a key characteristic of the MMN response [26]. In contrast, the N400 is– as noted–seemingly insensitive to pitch prediction violations. The results of Besson and Macar [41] and others [42–44, 61] consistently report an absence of N400 effects for musical mis-matches, instead most consistently reporting positive P300 effects. Crucially, such research has also demonstrated the strong, temporally-overlapping, positive P300 response evident in such conditions does not simply mask the N400 activity, but instead appears to occur instead of it [44]. It is important here to mention that a number of more recent papers have reported N400 responses to musical mismatches [62, 63], but that these consistently only occur when partici-pants are presented with familiar melodies simultaneously to words. That is, participants are actively processing both lexical and tonal input when the violation occurs. Furthermore, future research should be conducted–if possible—to determine whether the PMN responds to non-linguistic stimuli that the N400 appears to be sensitive to (i.e., pictures [64], gestures–[65], arithmetic [66], and environmental sounds [67].

The results of the present study therefore reflect a pattern of effects indicative of a compo-nent that may be selectively sensitive to linguistic manipulations, but crucially appears to respond very similarly to what we might expect for early N400 activity. The findings of the present study thus lead us to conclusions that are twofold. Firstly, as noted, the phoneme mis-match response (however one wishes to label the effect) appears language-specific. Such a find-ing replicates a great deal of work demonstrating the seeming language-specificity of ERP activity within the PMN/N400 time window. However, despite this finding, it is crucial that the field exercises caution in interpretations of the PMN as a *phonology*-specific response until further evidence can conclusively determine this. Whilst neurocognitive language science con-tinues to utilize the PMN as a dependent measure in theoretical research, despite the critical paucity in evidence to confirm the response's sensitivity to non-phonological manipulations (i.e., lexicality), the reliability of an ever-increasing body of research must be called into ques-tion. Albeit a significant problem within academic research, the implications within clinical settings, in which PMN paradigms have recently been adopted to shed light on language impairment [68–70], are perhaps greater still. Such research, which uses the supposed particu-lar sensitivities of the PMN and N400 to distinguish between specific impairments in

phonological processing or language comprehension, is at risk of producing findings that are misled by the general consensus of a well-established, reliable component with specific sensitivity to pre-lexical phonology. Secondly, our findings leave open the possibility that the PMN may either share considerable underlying mechanisms with, or in fact represent an early N400 effect response. Whilst further research is clearly necessary, it is also crucial that the field begin to consider the ever-increasing evidence for clear functional (and, potentially, neurobiological) links between the PMN and N400. Corroborating this is the topography of the phoneme mismatch effect of the present study, contributing to a growing body of research that points towards predominantly centro-parietal 'PMN' effects and, a clear overlap with characteristics typical of an N400 response. To sum, whilst the results of the present study clearly indicate a response that is language-specific, the selective sensitivity of the PMN to phonological manipulations necessitates further evidence, and revisitation by the field to longstanding distinctions assumed between PMN and early N400 sensitivities is crucial.

## Supporting information

**S1 File. Participant demographic information.**
(XLSX)

**S2 File. Full stimuli list and recordings.**
(ZIP)

**S3 File. Topography plots for the individual conditions for time periods denoted by the significant clusters.**
(DOCX)

## Author Contributions

**Conceptualization:** Jen Lewendon, Stephen Politzer-Ahles.

**Data curation:** Jen Lewendon, James Britton.

**Formal analysis:** Jen Lewendon, Stephen Politzer-Ahles.

**Funding acquisition:** Jen Lewendon, Stephen Politzer-Ahles.

**Investigation:** Jen Lewendon.

**Methodology:** Jen Lewendon, Stephen Politzer-Ahles.

**Project administration:** Jen Lewendon, James Britton.

**Supervision:** Stephen Politzer-Ahles.

**Visualization:** Jen Lewendon.

**Writing – original draft:** Jen Lewendon.

**Writing – review & editing:** Jen Lewendon, James Britton, Stephen Politzer-Ahles.

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
