## [Decision Letter · Decision Letter 0]

4 Sep 2023

PONE-D-23-09694The Phonological Mapping Negativity (PMN) as a language-specific component: exploring responses to linguistic vs musical mismatch.PLOS ONE

Dear Dr. Lewendon,

Thank you for submitting your manuscript to PLOS ONE. After careful consideration, we feel that it has merit but does not fully meet PLOS ONE’s publication criteria as it currently stands. Therefore, we invite you to submit a revised version of the manuscript that addresses the points raised during the review process.

We look forward to receiving your revised manuscript.

Kind regards,

Sergei Tugin, PhD

Academic Editor

PLOS ONE

Journal Requirements:

2. Please ensure that you refer to Figure 4 in your text as, if accepted, production will need this reference to link the reader to the figure.

Reviewers' comments:

Reviewer's Responses to Questions

**Comments to the Author**

1. Is the manuscript technically sound, and do the data support the conclusions?

Reviewer #1: Partly

Reviewer #2: Yes

2. Has the statistical analysis been performed appropriately and rigorously? 

Reviewer #1: Yes

Reviewer #2: Yes

3. Have the authors made all data underlying the findings in their manuscript fully available?

Reviewer #1: Yes

Reviewer #2: Yes

4. Is the manuscript presented in an intelligible fashion and written in standard English?

Reviewer #1: Yes

Reviewer #2: Yes

5. Review Comments to the Author

Reviewer #1: The article addresses a theoretically well motivated question on the field of the electrophysiological study of linguistic processes: is the Phonological Mismatch Negativity (PMN) ERP component language specific? Moreover, is the PMN distinct from other similar ERP components such as the MMN and N400? The authors claim that the results of the study provides a definitive answer to the first question, but leave the answer to the second question to further studies. I appreciate the importance of the research question investigated in the study, but I have some doubts whether the methodology applied could in fact provide decisive answers to this question.

The article is a concise and well written summary of the study, with some minor typos (see later). The introduction presents a compelling summary of the theoretical background of the PMN component and clearly highlights the theoretical and methodological challenges related to this ERP component. The authors propose a new experimental paradigm to investigate the language specificity of the PMN. This paradigm is markedly different from the paradigms used previously to elicit the PMN. The authors present a lengthy analysis in the introduction about the controversies with the previous paradigms and their reliability to elicit the PMN. I think this is a particularly valuable section of the article. In the following, I present my comments on the article separated to major and minor issues.

Major issue

Experimental paradigm. As I mentioned previously, I have doubts if the paradigm used can provide reliable results related to the linguistic nature of the PMN. My concern is the simultaneous presentation of the linguistic and non-linguistic stimuli. While I think that this a clever way to control for confounds present in previous experiments, I wonder if the ERPs to a complex simultaneous stimulus aggregate are comparable to the ERPs obtained as a correlate of purely linguistic information. I think the authors need to provide a more convincing argument that the results of the present study are comparable to the results of previous studies. Furthermore, even if attention is directed to only one information (either speech of notes), there might be a pre-attentive processing of the non-attended information, which is in fact a way to elicit the MMN. How can the authors make sure that this pre-attentive processing is not manifested in the ERPs somehow?

Minor issues

1. Introduction: I miss from the introduction a more detailed functional characterisation of the PMN component. The authors mention the PMN “indexes phonological processing at the pre-lexical level”, but this is a very broad description. What does phonological processing mean? What are the input-output processes involved here? I think the article would benefit from a more thorough explanation of the possible functions of PMN in the Introduction, and could help the authors to situate the PMN in the Bornkessel-Schlesewsky and Schlesewsky account in the Discussion.

2. Participants: out of 52 participants, 20 were excluded because of low accuracy in the task. This is a huge number of participants excluded, and raises a questions about the reliability of the paradigm, i.e., if the task was too difficult for a large number of participants. I think the authors should discuss this aspect of the study. Furthermore, some of the participants were left handed. Given that the hemispheric localisation of language related brain areas is unknown in left handed participants, I think it’s not a good idea to include them in the sample.

3. Figure 1.: While the figure is clear and informative, I wonder if the order of presentation of the target and prime could be changed to better reflect the temporal relations of the two stimuli? I.e., the prime is followed by the target, not the other way around.

4. Data pre-processing: The authors applied a low-pass filter of 30 Hz on the segmented data as a final step during the pre-processing. My understanding is that filtering is more reliable when applied on continuous data. Could the authors explain the rationale why not using the low-pass filter together with the high-pass filter?

Typos

p2.para3: “disassociating the PMN from= from the ERP” - ?

p3. para1: “other processing mechanisms Despite considerable “ - missing “.”

p4. para1: “on the PMN amplitude `cannot” - ‘

p6. para1: “featured note which were either correct” - notes

p6. para2: "the presence of a PMN effect for linguistic manipulations in Besson and Macar cannot be conclusively determined” - Besson and Macar study?

p8. para2: “sound-calling earphones” - maybe noise cancelling? why is this important? Can you specify the type of earphones used?

p13. para2: “presented with familiar melodies simultaneously to word” - ?

Reviewer #2: SUMMARY:

The study investigates the nature of the Phonological Mismatch Negativity (PMN), in particular its language-specificity, and thereby by proxy its relation to the N400 and the MMN components.

This is done using a phoneme-deletion paradigm (which has previously been used for investigating the PMN) in combination with a parallel note-deletion (musical) paradigm.

The study finds support for language-specificity of the PMN via significant negative deflections in the phonological mismatch condition, but not in the musical mismatch condition.

GENERAL COMMENTS:

I wish to congratulate the authors on a great paper. The study is neatly and concisely motivated in the introduction. The paper is clearly and well written throughout. The (full) study itself has an elegant design and the results (that are included) are clearly reported. I also greatly appreciate the supplementary material.

I do, however, have two main concerns that I'd like to see addressed before I can recommend publication.

MAIN CONCERN(S):

1. First off, the comprehensive introduction introduces and motivates the PMN as a component that could potentially be independent of both the MMN and the N400. And the elegant design of the study even supports this endeavour of disentangling the PMN from the two other components, but then only the "speech-specific" aspect of the design is addressed in this study - the attentional aspect is completely ignored (albeit directly integrated in the design). In the Methods section, one can then deduce that the attentional aspect is addressed in a parallel paper. I must admit I struggled quite a bit to come to terms with this decision - and I'm not sure I fully have come to terms with it. There are so many good reasons for integrating the two papers into one (especially since the results from the other paper can't be compared to this one due to radically differences in preprocessing/analysis, i.e. number of participants included). My suggestion is therefore to integrate the attentional contrast in this study as well and just clearly disclaim that those data have already been published (together with another experiment). It would make the integration and interplay of the premise and design of the study stand so much stronger, and thus the conclusions would form a much more coherent picture.

2. The fact that 20 (out of 52 - or 54? - participants) scored below 50% accuracy seems quite concerning - especially since this number was much lower in the parallel study where only the phonemic mismatches were considered. It tentatively suggests that the tonal mismatch part of the task was harder than the phonemic one (which seems a bit counter-intuitive to me after having listened to the stimuli). Under all circumstances, I strongly urge you to do an by-item analysis on the behavioral results as well to see whether that provides any relevant insight (e.g. if certain match/mismatch items were particularly hard to identify, some of the correct responses for those items in the included participants may not be fully credible)

And then conduct a supplementary analysis with no participants or trials rejected on the basis of incorrect responses to see if and how much that actually matters for the results.

On the note of the by-item analysis, in the Methods it says "The rationale for reducing note duration was to ensure that the processing of phonological input was not impeded by the tones." > But since the onset of the syllables are the only distinguishable cue for the phoneme-mismatches, is it not a bit unfortunate that the tones and syllables onset simultaneously? My impression when listening to the stimuli was this somewhat impedes the processing of the very rapid transition cues in the onsets of the syllables - I could even imagine this being particularly difficult for specific sounds, hence an even greater need for a by-item analysis of the behavioral responses. I don't think, however, that the joint onsets could be the root of the potentially more difficult tonal condition - since the notes are sustained for longer periods of time than the syllable onsets.

3. "Within each block 10 items formed the present study..." > Plz explain the procedure/block structure much more clearly, incl. using tables and the like. I've tried very hard to parse this passage (while conferring with the procedure description and tables in other paper), and I still can't fully grasp the numbers and division of conditions between the two papers. Here's where I'm at currently:

The LCN paper made use of the following conditions:

[Attention: Phoneme] > [correct note, correct phoneme] and [correct note, incorrect phoneme]

[Attention: Tune] > [correct note, correct phoneme] and [correct note, incorrect phoneme]

(and thus for both attention modes, did not make use of [incorrect note, correct phoneme] and [incorrect note, incorrect phoneme]

This paper made use of the following conditions:

[Attention: Phoneme] > [correct note, correct phoneme], [correct note, INCORRECT PHONEME] and [incorrect note, incorrect phoneme]

[Attention: Tune] > [correct note, correct phoneme], [INCORRECT NOTE, correct phoneme] and [incorrect note, incorrect phoneme]

(and thus for phoneme and tune attention modes, did not make use of either [incorrect note, correct phoneme] or [incorrect note, correct phoneme], respectively

This can be gathered from this section in the Methods:

"The nature of the design meant that for counterbalancing purposes, participants experienced two additional conditions that were not of relevance to the present study (correct note & incorrect phoneme in the blocks in which participants focussed on the tune, and incorrect note & correct phoneme in the blocks in which participants were directed to attend to the word."

However, in a separate part of the Methods:

"Within each block 10 items formed the present study (i.e, the conditions correct note & correct phoneme, correct note & incorrect, incorrect note & correct phoneme, incorrect note & incorrect phoneme), whilst 10 formed a separate study not reported here."

I see how 10 (out of 20) items were used in the LCN paper cuz they ignored half of the conditions (which would be good to also explicate here, i.e. which conditions were used in that study), but I don't see how that leaves only 10 for this study which only ignored 1/4 of the conditions (namely the [correct note & incorrect phoneme] in the Phoneme-attention mode and the [incorrect note & correct phoneme] in the Tune-attention mode)?

Furthermore, the fact that all four conditions are mentioned in the i.e.-parenthesis makes it rather hard to remember/discern that for the [correct note & incorrect phoneme] and [incorrect note & correct phoneme] this only pertains to the "relevant" attention mode.

In addressing my concerns in point 1 (and partly point 2), the reporting of the design may change slightly, but it would still require a much more clearer explication and visual presentation to be accessible to the readers.

I will re-assess the Discussion once the above points have been addressed.

MINOR:

FIGURES:

Figure 2 & 3 captions: topographies are mentioned but not included - I'm assuming they have just been moved to the bottom of Figure 4 and then the captions haven't been updated (or the other way around)?

No line or page numbers provided, hence I can only refer to what main sections of the text a given change relates to:

INTRODUCTION:

"PMN from= from the" > "PMN from the"

"(Van den brink & Hagoort, 2004; Van den Brink et al., 2001; Hagoort & Brown, 2000); Connolly & Phillips, 1994)" > (Van den brink & Hagoort, 2004; Van den Brink et al., 2001; Hagoort & Brown, 2000; Connolly & Phillips, 1994) [i.e. delete the extra closing parenthesis]

"phonologically incongruent" (or "phonological incongruence") > I'm thinking "unexpected" or something similar is better suited here - to me, phonologically incongruent would rather be something like "*shand" in stead of "hand"

"In their 2020 review, Lewendon et al." > Since the first author overlaps with the current study, it sounds peculiar to say "their" here, consider rephrasing to acknowledge the link to the current author list more clearly

"PMN amplitude `cannot necessarily be deemed meaningful" > "PMN amplitude cannot necessarily be deemed meaningful" [delete the accent aigu]

""Lewendon, Britton & Politzer-Ahles (under review/forthcoming)" > has now been published and thus needs updating

"this shared functional sensitivity, alongside overlapping topography the timing invite questions" > I can't parse this properly - there's probably a comma missing after "topography", but I still need another word or two somewhere before or after "the timing" to make proper sense of the sentence

"featured note which were either correct" > "featured notes which ..."

"ERP component is selectively sensitivity to linguistic manipulations" > "selectively sensitive ..."

"the mechanisms that given rise to" > "have given" or "gave"

"In contrast, the MMN has been reliably demonstrated ..." > plz consider also including some of the papers on musical multi feature paradigms, e.g.

https://pubmed.ncbi.nlm.nih.gov/21621766/

https://www.ncbi.nlm.nih.gov/pmc/articles/PMC6990974/

"word (e.g., Prime: /pɪˈrɑːnə/)" > none of the other examples are accompanied by their labels (e.g. Target), so I suggest opting for consistency in that regard, and then perhaps also spell out the word - not all readers of PLoS ONE can be expected to be fully fluent in parsing phonemic transscriptions.

"Directing participant attention to either the notes or linguistic stimuli, we therefore independently manipulated either tone or phoneme match/mismatch, whilst retaining physically identical target stimuli across conditions." > since there aren't any objective measures of directed attention in this setup AND because the performance was so relatively poor in many participants, I would suggest to rephrase this sentence in a less categorical manner - I find it a bit of a push to say that tone or phoneme match/mismatch were "independently manipulated" due to the direction of participant attention.

METHODS:

"Fifty-two native English speakers took part in the experiment." > In the other paper on the same dataset, 54 initial participants are mentioned - in this one, only 52. Why the discrepancy?

"(iv) incorrect note & incorrect phoneme (e.g., ambitious – bitious; A4-B4-C5 – E5-C5)" > "licious"

"were directed to attend to the word" > "were directed to attend to the word)" [i.e. close the parenthesis]

"Note stimuli were in Sibelius®" > "were recorded/generated in Sibelius®"?

"To The mean intensity of each" > "The mean intensity of each"

"(i.e, the conditions correct note & correct phoneme, correct note & incorrect," > "correct note & incorrect phoneme,"

"data was low-pass filtered 30 Hz" > "filtered at 30 Hz"

"A total of 78 trials were removed during artifact rejection" > could the order of the artefact rejection plz be switched. That way the numbers for the incorrect responses reflect how many trials were rejected for the different conditions on that particular basis (to better reflect potential differences between conditions using that criterion), and then the extreme value-based rejection routine can be applied subsequently and a summary of how many trials were left for each condition for statistical analyses can be given.

"leaving an average of 38.69/40 trials" > plz provide SD across participants for these averages

RESULTS:

"Overall accuracy was 94% in the phoneme match condition, 99% in the tone match condition, 77% in the tone mismatch condition, and 81% in the phoneme mismatch condition." > I'm assuming that these numbers don't align with the numbers mentioned in the artefact routine because the extreme-value based rejection was run first, right?

"The increase in negativity was driven by a cluster occurring approximately 387 - 500 ms across central, parietal and occipital electrodes" > plz make explicit that the proposition "was driven" is based on visual inspection (I presume), and then make a note of the inherent limitations of spatial and temporal conclusions made based on cluster-based permutation tests with a reference to the Sassenhagen & Drasckow paper (https://onlinelibrary.wiley.com/doi/10.1111/psyp.13335) and perhaps the fieldtrip-note as well (https://www.fieldtriptoolbox.org/faq/how_not_to_interpret_results_from_a_cluster-based_permutation_test/). This cautious wording should then also be applied to the remainder of the temporal and spatial interpretations made in the Results section - incl. the caption to Figure 4 "electrodes labels in black denote electrodes that formed part of the cluster that drives the significant difference between conditions".

"across central, parietal and occipital electrodes (Figure 1A)" > "(Figure 4A)"

"more positive response than tone match (p = 0.019) (Figure 1B)" > "(Figure 4B)"

SUPPLEMENTARY MATERIAL:

Again, huge kudos for providing this material. Really a great help in properly understanding the study.

In the audio files, there are 20 primes (incl. targets) that are not listed in stimulus spreadsheet (and which weren't paired with any incorrect note versions) - what was the role of these stimuli?

atlantic

competing

complaining

completion

computer

condensing

detective

director

enquiry

eviction

exceeding

exciting

extinction

fixation

impressing

inspection

pacific

quotation

umbrella

withdrawal

PRACTICAL COMMENTS:

Plz provide line (and page) numbers in the revised version of the manuscript

6. PLOS authors have the option to publish the peer review history of their article (what does this mean?). If published, this will include your full peer review and any attached files.

Reviewer #1: No

Reviewer #2: **Yes: **Andreas Højlund

---

## [Author Response · Author response to Decision Letter 0]

6 Nov 2023

Please see attached reviewer response letter for formatted response.

2. Please ensure that you refer to Figure 4 in your text as, if accepted, production will need this reference to link the reader to the figure.

Point-by-point response

5. Review Comments to the Author

Reviewer #1: The article addresses a theoretically well motivated question on the field of the electrophysiological study of linguistic processes: is the Phonological Mismatch Negativity (PMN) ERP component language specific? Moreover, is the PMN distinct from other similar ERP components such as the MMN and N400? The authors claim that the results of the study provides a definitive answer to the first question, but leave the answer to the second question to further studies. I appreciate the importance of the research question investigated in the study, but I have some doubts whether the methodology applied could in fact provide decisive answers to this question.

The article is a concise and well written summary of the study, with some minor typos (see later). The introduction presents a compelling summary of the theoretical background of the PMN component and clearly highlights the theoretical and methodological challenges related to this ERP component. The authors propose a new experimental paradigm to investigate the language specificity of the PMN. This paradigm is markedly different from the paradigms used previously to elicit the PMN. The authors present a lengthy analysis in the introduction about the controversies with the previous paradigms and their reliability to elicit the PMN. I think this is a particularly valuable section of the article. In the following, I present my comments on the article separated to major and minor issues.

General response: We thank the reviewer for the time they have afforded our submission and the constructive comments and suggestions they offer. We particularly appreciate the depth with which the reviewer has considered the methodology of our paper and its overall aims, alongside proposed amendments which we feel have notably improved the manuscript. We have sought to implement the suggested changes where possible, or respond to points in a manner that provides further clarity. We hope that our revisions to each point, laid out below, satisfy the reviewer’s questions and concerns.

Major issue

Experimental paradigm. As I mentioned previously, I have doubts if the paradigm used can provide reliable results related to the linguistic nature of the PMN. My concern is the simultaneous presentation of the linguistic and non-linguistic stimuli. While I think that this a clever way to control for confounds present in previous experiments, I wonder if the ERPs to a complex simultaneous stimulus aggregate are comparable to the ERPs obtained as a correlate of purely linguistic information. I think the authors need to provide a more convincing argument that the results of the present study are comparable to the results of previous studies. Furthermore, even if attention is directed to only one information (either speech of notes), there might be a pre-attentive processing of the non-attended information, which is in fact a way to elicit the MMN. How can the authors make sure that this pre-attentive processing is not manifested in the ERPs somehow?

Response: We acknowledge and appreciate the theoretical grounding of these questions, but we are unsure that the data we present support the points made. The reviewer’s first concern is that ERP responses to simultaneously presented tone and phoneme mismatches might not be comparable to simple phoneme matches. We agree that this is a legitimate question, and would warrant consideration of the validity of our findings had we not observed a reliable PMN response to the phoneme mismatches. Granted, we make a point of observing that the PMN effect we report is notably centro-parietal (as compared to the more typical characterization of the response as fronto-central), but given that there is widespread evidence of centro-parietal PMN effects in literature manipulating simple phoneme match/mismatch (noted in the text), this possibility - that our paradigm might not have succeeded in producing a PMN response that is sufficiently comparable to other literature - doesn’t seem like a criticism that is supported by our findings. From a slightly more theoretical perspective, the point could also perhaps be made that all linguistic stimuli are multidimensional, so a spoken word, for example, is in itself a “complex stimulus aggregate” (consisting of many phonetic features overlapping) just as our stimuli are.

In the latter half of this concern, the reviewer suggests that pre-attentive processing of the non-target stimuli might influence the ‘tone only’ or ‘phoneme only’ conditions. Again, this seems unlikely given the results of a previously published paper (https://people.ku.edu/~sjpa/pubs/Lewendonetal2023.pdf) in which we establish that eliciting the PMN in pre-attentive contexts does not appear possible. Furthermore, it should be noted that the present experimental paradigm didn’t have the oddball ratio necessary to elicit MMN responses. 

Given the above, and the implications of using a design in which stimuli are not comparable across conditions, it feels to us that although presenting phoneme and tone stimuli concurrently might come with its own limitations, not doing so would result in a notably less well-controlled paradigm in which any effects might be attributed to a host of differences in perceptual qualities of the stimuli.

Minor issues

1. Introduction: I miss from the introduction a more detailed functional characterisation of the PMN component. The authors mention the PMN “indexes phonological processing at the pre-lexical level”, but this is a very broad description. What does phonological processing mean? What are the input-output processes involved here? I think the article would benefit from a more thorough explanation of the possible functions of PMN in the Introduction, and could help the authors to situate the PMN in the Bornkessel-Schlesewsky and Schlesewsky account in the Discussion.

Response: One of the biggest problems with the PMN is that it is a notably poorly defined and poorly understood component. Here we quote the original characterizations of the PMN (‘phonological processing at the pre-lexical level’) as it was/has been explained in past literature, but go on to highlight the fact that- in reality the effect is badly documented and understood both in terms of its spatiotemporal characteristics and functional significance. It is difficult to elaborate on a characterization of the component produced by others, and that - through the course of the paper - we note the opacity of. However, this comment has been particularly helpful in ensuring we emphasize the lack of consistent characterization of the PMN early on – and we have amended the language of the characterization (P2, lines 57-58; 62; 66) to make it clearer that we approach this characterization with caution. 

2. Participants: out of 52 participants, 20 were excluded because of low accuracy in the task. This is a huge number of participants excluded, and raises a questions about the reliability of the paradigm, i.e., if the task was too difficult for a large number of participants. I think the authors should discuss this aspect of the study. Furthermore, some of the participants were left handed. Given that the hemispheric localisation of language related brain areas is unknown in left handed participants, I think it’s not a good idea to include them in the sample.

Response: First, accuracy. We do think that the task was more difficult than anticipated, and this isn’t something we would dispute at all. However, we don’t necessarily feel this implies that the paradigm is not reliable, nor does it render the results redundant. Hypothetically, if we were to assume the paradigm was indeed entirely too difficult to elicit mismatch responses, this should manifest as a complete lack of responses to both phoneme and tone mismatch. Given that reliable responses were found, it seems reasonable to assume although the paradigm was more difficult than we anticipated, participants still detected both types of mismatch. However, based on both this comment we have rerun the analysis including:

1. Only the high accuracy participants and only correct trials (note that the accuracy thresholds differ from those reported in the original paper); 

Note mismatch - positive effect (p = .073)

 - negative effect (NS)

Phoneme mismatch - positive effect (NS)

 - negative effect (p = .049) *

2. both the high and low accuracy participants (correct only trials)

Note mismatch - positive effect (p = .032) *

 - negative effect (NS)

Phoneme mismatch - positive effect (NS)

 - negative effect (p = .015)*

3. High and low accuracy participants with both correct and incorrect trials included (see full paper).

Note mismatch - positive effect (p = .028) *

 - negative effect (NS)

Phoneme mismatch - positive effect (NS)

 - negative effect (p = .008) *

Because of the manner in which the effects pan out, we attribute the retention in the trend of effects, but loss of significance across exclusions as an effect of reduced power. Given that the same trends appear across the data, and that exclusion of the high proportion of participants caused concern for both Reviewer 1 and 2, we have instead decided to report results for all participants, irrespective of accuracy, with a footnote to mention that when excluding the low-accuracy participants and/or incorrect trials, and/or left-handed participants, the same general qualitative pattern of results was obtained but didn’t always reach statistical significance.

3. Figure 1.: While the figure is clear and informative, I wonder if the order of presentation of the target and prime could be changed to better reflect the temporal relations of the two stimuli? I.e., the prime is followed by the target, not the other way around.

Response: We agree. To try and better convey order of presentation we have amended this figure sure that the prime is above the target.

4. Data pre-processing: The authors applied a low-pass filter of 30 Hz on the segmented data as a final step during the pre-processing. My understanding is that filtering is more reliable when applied on continuous data. Could the authors explain the rationale why not using the low-pass filter together with the high-pass filter?

Response: We have rerun the preprocessing to move the low-pass filter prior to segmentation. All results and figures now reflect this change, and lines 365-370 (p8) have been updated accordingly.

Typos

p2.para3: “disassociating the PMN from= from the ERP” - ?

p3. para1: “other processing mechanisms Despite considerable “ - missing “.”

p4. para1: “on the PMN amplitude `cannot” - ‘

p6. para1: “featured note which were either correct” - notes

p6. para2: "the presence of a PMN effect for linguistic manipulations in Besson and Macar cannot be conclusively determined” - Besson and Macar study?

p8. para2: “sound-calling earphones” - maybe noise cancelling? why is this important? Can you specify the type of earphones used?

p13. para2: “presented with familiar melodies simultaneously to word” - ?

Response: Thank you!

Reviewer #2: SUMMARY:

The study investigates the nature of the Phonological Mismatch Negativity (PMN), in particular its language-specificity, and thereby by proxy its relation to the N400 and the MMN components.

This is done using a phoneme-deletion paradigm (which has previously been used for investigating the PMN) in combination with a parallel note-deletion (musical) paradigm.

The study finds support for language-specificity of the PMN via significant negative deflections in the phonological mismatch condition, but not in the musical mismatch condition.

GENERAL COMMENTS:

I wish to congratulate the authors on a great paper. The study is neatly and concisely motivated in the introduction. The paper is clearly and well written throughout. The (full) study itself has an elegant design and the results (that are included) are clearly reported. I also greatly appreciate the supplementary material.

I do, however, have two main concerns that I'd like to see addressed before I can recommend publication.

General response: We first would like to thank the reviewer for the time and consideration they have afforded our paper, and really appreciate their kind words regarding the paradigm and design. The level of attention the reviewer has given to our manuscript is incredibly apparent in the detailed and comprehensive points, corrections and comments, both major and minor. The suggestions provided are insightful and have – wherever possible – been fully implemented to the overall benefit of the paper. Where this has not been feasible, we have tried our best to outline the rationale to the reviewer and make any amendments to the paper that reflect any concerns. At times the comments of Reviewer 2 overlap with those of Reviewer 1, but result in different suggestions in terms of revisions proposed to resolve these issues. Where this has arisen, we have attempted to revise analyses to address the underlying concerns, albeit varying in whether the implementation of suggestions sways more towards the recommendations of one reviewer over the other.

MAIN CONCERN(S):

1. First off, the comprehensive introduction introduces and motivates the PMN as a component that could potentially be independent of both the MMN and the N400. And the elegant design of the study even supports this endeavour of disentangling the PMN from the two other components, but then only the "speech-specific" aspect of the design is addressed in this study - the attentional aspect is completely ignored (albeit directly integrated in the design). In the Methods section, one can then deduce that the attentional aspect is addressed in a parallel paper. I must admit I struggled quite a bit to come to terms with this decision - and I'm not sure I fully have come to terms with it. There are so many good reasons for integrating the two papers into one (especially since the results from the other paper can't be compared to this one due to radically differences in preprocessing/analysis, i.e. number of participants included). My suggestion is therefore to integrate the attentional contrast in this study as well and just clearly disclaim that those data have already been published (together with another experiment). It would make the integration and interplay of the premise and design of the study stand so much stronger, and thus the conclusions would form a much more coherent picture.

We acknowledge this point and feel that it is a valid question to ask regarding the presentation of our findings. Our rationale for splitting the results into separate papers was predominantly driven by an attempt not to muddy the waters. The question of whether the PMN is distinguishable from the N400 has received quite some attention over the years. On the other hand, whether the PMN might result from the same underlying mechanism as the MMN is far less represented/tested/questioned (in fact we’re not aware of any other papers apart ours - which forms the inattentive part of this paradigm – that ask this). Instead, the independence of these two components seem to be simply assumed and accepted. 

Splitting our results into two separate papers allowed us to fully hone in on the key functional sensitivities of the PMN relied upon to propose its independence from t

---

## [Decision Letter · Decision Letter 1]

20 Feb 2024

PONE-D-23-09694R1The Phonological Mapping Negativity (PMN) as a language-specific component: exploring responses to linguistic vs musical mismatch.PLOS ONE

Dear Dr. Lewendon,

Thank you for submitting your manuscript to PLOS ONE. After careful consideration, we feel that it has merit but does not fully meet PLOS ONE’s publication criteria as it currently stands. Therefore, we invite you to submit a revised version of the manuscript that addresses the points raised during the review process.

We look forward to receiving your revised manuscript.

Kind regards,

Laura Morett

Academic Editor

PLOS ONE

Journal Requirements:

Additional Editor Comments:

I thank the authors for their attention to the reviewers' feedback. I have reviewed the revised manuscript and agree with the reviewers that the authors have been responsive to their concerns. R2 raises a few additional comments that should be addressed prior to publication, so I encourage the authors to do so. I look forward to receiving another revision addressing these remaining comments.

Reviewers' comments:

Reviewer's Responses to Questions

**Comments to the Author**

1. If the authors have adequately addressed your comments raised in a previous round of review and you feel that this manuscript is now acceptable for publication, you may indicate that here to bypass the “Comments to the Author” section, enter your conflict of interest statement in the “Confidential to Editor” section, and submit your "Accept" recommendation.

Reviewer #1: All comments have been addressed

Reviewer #2: All comments have been addressed

2. Is the manuscript technically sound, and do the data support the conclusions?

Reviewer #1: (No Response)

Reviewer #2: Yes

3. Has the statistical analysis been performed appropriately and rigorously? 

Reviewer #1: (No Response)

Reviewer #2: Yes

4. Have the authors made all data underlying the findings in their manuscript fully available?

Reviewer #1: (No Response)

Reviewer #2: Yes

5. Is the manuscript presented in an intelligible fashion and written in standard English?

Reviewer #1: (No Response)

Reviewer #2: Yes

6. Review Comments to the Author

Reviewer #1: (No Response)

Reviewer #2: GENERAL COMMENTS:

This time around I wish to congratulate the authors on a tremendous effort in addressing all the concerns raised by the other reviewer and myself. Data have been re-analyzed (by also incl. the low-accuracy participants - and the preprocessing has also been changed according to my recommendation).

The result is a much improved paper. Clarity wrt design and stimuli has also improved greatly.

I only have a few minor concerns that I recommend be addressed before I can recommend publication.

MINOR CONCERNS:

PROOFREADING:

During the revision process quite a few typos and the like have crept in, so it seems an extra proofreading is warranted, e.g.:

396: "in negativity most pronounced" > "in negativity [was] most pronounced"

Footnote 3: "didn't" > "did not"

499: "gestures – [65]" > "gestures [65]"

TABLE 2:

Great addition, thank you.

Regarding readability of the table, I would suggest including a space between P± and N± (and even better, fully aligning all P± and all N±, resp - e.g. via split cells or however, the authors might prefer to do this). For the Y and N column (aka. "Incl. in present study"), I would suggest removing all N and only include Y (as they are listed now, Y and N are quite difficult to visually distinguish, hence either leaving all N blanks, or denote with a hyphen (-), would greatly improve readability).

FIGURES 2-3:

I acknowledge that the authors have chosen to mainly delete references to the temporal extent of the main cluster identified in the statistical analyses in the main text in order to address the concerns I raised with reporting the temporal (and spatial) extent of clusters from cluster-based permutation testing; however, I find that the reader is left at a disadvantage in interpreting the myriad of waveforms (12 per figure) when there's no temporal guidance. Hence, the vertical bars from the original plots could perhaps be re-introduced with a reference to a similar formulation as in Fig 4 (as well as to the raster plots and topographies in Fig 4). And finally I again urge the authors to just include a one-liner mentioning caution in interpreting cluster extent too strongly/directly in cluster-based permutation testing (with a reference to Sassenhagen & Drasckow (2019)).

DISCUSSION:

As stated in my first review, I can now assess the Discussion more clearly, given the final results of the revision.

I generally applaud the critical reading of the literature in the Discussion in combination with the contextualization of the results of the present study.

However, I do find that the first section of the Discussion on the topographies (ll 436-445) leaves the reader a little short-handed.

First off, it's hard to see how "Both effects clearly show a frontal positive and posterior negative pattern" - to me, the PMN pattern is much more lateralized than frontal-posterior, so it would be great to explicate this proposed similarity in a way that more clearly guides the reader towards what the authors have in mind.

Next, without any supporting references, the generalization that "Whilst such an inversion of effects – with components consisting of opposite polarities on one side of the head compared to the other is typical of all ERP responses, the similar dipoles but slightly different orientations of the topography plots perhaps warrant further investigation" seems a bit hasty... this is certainly true of MEG data (from magnetometers), but less so of EEG data, esp. if we assume that (at least part of) the observed responses originate in the auditory cortices (then the opposite polarity will often not be detectable cuz we cannot record EEG from the below the brain).

Further, the authors themselves refer later in the Discussion to work on musical mismatches (ll 490-492) that elicit frontal P300-like negativities. This seems very relevant to bring into play here - in order to better understand the frontal positivity elicited in the note condition. Is this most sensibly interpreted as an early P300/P3a?

Finally, how common is this right-posterior negativity of the PMN compared to the literature. How much of this can be related to the concurrent presentation of the musical stimuli? Any literature on how this concurrent presentation affects the topography of these early responses?

Hence, I urge the authors to substantiate the discussion of the differences in topographies considerably (especially considering the above-mentioned references later in the discussion). And then I urge the authors to anchor the speculation regarding "similar dipoles but slightly different orientations of the topography" in some literature in some way - either generally wrt the plausibility of the similarity of the underlying dipoles given the topographies or specifically wrt previous literature on almost identical paradigms.

In all of the above, it's important to bear in mind that the interpreted topographies reflect difference waveforms - and thus, any dipole inferences hinges on the topographies reflecting an underlying "difference/mismatch" response and not just differences in different co-occurring responses. Hence, to guide the reader in all of the above, it would be of great help if the topographies of the underlying responses to the match and mismatch conditions for the reported time window together with the rather clear P1-N1-P2 complex could be visualized either in the main paper or in the supplementary material for reference?

REFERENCES:

Sassenhagen J, Draschkow D (2019). Cluster‐based permutation tests of MEG/EEG data do not establish significance of effect latency or location. Psychophysiology. e13335. https://doi.org/10.1111/psyp.13335

7. PLOS authors have the option to publish the peer review history of their article (what does this mean?). If published, this will include your full peer review and any attached files.

Reviewer #1: **Yes: **Ferenc Honbolygó

Reviewer #2: **Yes: **Andreas Højlund

---

## [Author Response · Author response to Decision Letter 1]

24 Oct 2024

Reviewer #2: GENERAL COMMENTS:

This time around I wish to congratulate the authors on a tremendous effort in addressing all the concerns raised by the other reviewer and myself. Data have been re-analyzed (by also incl. the low-accuracy participants - and the preprocessing has also been changed according to my recommendation).

The result is a much improved paper. Clarity wrt design and stimuli has also improved greatly.

I only have a few minor concerns that I recommend be addressed before I can recommend publication.

We would like to thank the reviewer once again for the time that he has taken suggesting improvements to our manuscript. A review of this depth, with its level of constructive and consideration is a no quick endeavor, and we are genuinely grateful to the reviewer for the fact that our paper has benefited from this process not just once but twice. We feel that all the changes he has suggested across the two revisions have strengthened the paper substantially, and thank him for the effort and energy he has expended meticulously going through our manuscript.

MINOR CONCERNS:

PROOFREADING:

During the revision process quite a few typos and the like have crept in, so it seems an extra proofreading is warranted, e.g.:

396: "in negativity most pronounced" > "in negativity [was] most pronounced"

Footnote 3: "didn't" > "did not"

499: "gestures – [65]" > "gestures [65]"

Thank you for noting these, we’ve made the corrections.

TABLE 2:

Great addition, thank you.

Regarding readability of the table, I would suggest including a space between P± and N± (and even better, fully aligning all P± and all N±, resp - e.g. via split cells or however, the authors might prefer to do this). For the Y and N column (aka. "Incl. in present study"), I would suggest removing all N and only include Y (as they are listed now, Y and N are quite difficult to visually distinguish, hence either leaving all N blanks, or denote with a hyphen (-), would greatly improve readability).

As recommended, we have split the condition column into two cells to improve the readability. We would ideally prefer to retain the N in the ‘incl. in present study’ column, but absolutely agree in terms of the readability. To address this, we have changed the font colour of the Ns to grey.

FIGURES 2-3:

I acknowledge that the authors have chosen to mainly delete references to the temporal extent of the main cluster identified in the statistical analyses in the main text in order to address the concerns I raised with reporting the temporal (and spatial) extent of clusters from cluster-based permutation testing; however, I find that the reader is left at a disadvantage in interpreting the myriad of waveforms (12 per figure) when there's no temporal guidance. Hence, the vertical bars from the original plots could perhaps be re-introduced with a reference to a similar formulation as in Fig 4 (as well as to the raster plots and topographies in Fig 4). And finally I again urge the authors to just include a one-liner mentioning caution in interpreting cluster extent too strongly/directly in cluster-based permutation testing (with a reference to Sassenhagen & Drasckow (2019)).

Thank you for this, we agree that the amendment to the figure combined with the reference (I’ve placed this directly underneath the cluster figures) will definitely make things clearer for the reader.

DISCUSSION:

As stated in my first review, I can now assess the Discussion more clearly, given the final results of the revision.

I generally applaud the critical reading of the literature in the Discussion in combination with the contextualization of the results of the present study.

However, I do find that the first section of the Discussion on the topographies (ll 436-445) leaves the reader a little short-handed.

First off, it's hard to see how "Both effects clearly show a frontal positive and posterior negative pattern" - to me, the PMN pattern is much more lateralized than frontal-posterior, so it would be great to explicate this proposed similarity in a way that more clearly guides the reader towards what the authors have in mind.

Next, without any supporting references, the generalization that "Whilst such an inversion of effects – with components consisting of opposite polarities on one side of the head compared to the other is typical of all ERP responses, the similar dipoles but slightly different orientations of the topography plots perhaps warrant further investigation" seems a bit hasty... this is certainly true of MEG data (from magnetometers), but less so of EEG data, esp. if we assume that (at least part of) the observed responses originate in the auditory cortices (then the opposite polarity will often not be detectable cuz we cannot record EEG from the below the brain).

Further, the authors themselves refer later in the Discussion to work on musical mismatches (ll 490-492) that elicit frontal P300-like negativities. This seems very relevant to bring into play here - in order to better understand the frontal positivity elicited in the note condition. Is this most sensibly interpreted as an early P300/P3a?

Finally, how common is this right-posterior negativity of the PMN compared to the literature. How much of this can be related to the concurrent presentation of the musical stimuli? Any literature on how this concurrent presentation affects the topography of these early responses?

Hence, I urge the authors to substantiate the discussion of the differences in topographies considerably (especially considering the above-mentioned references later in the discussion). And then I urge the authors to anchor the speculation regarding "similar dipoles but slightly different orientations of the topography" in some literature in some way - either generally wrt the plausibility of the similarity of the underlying dipoles given the topographies or specifically wrt previous literature on almost identical paradigms.

In all of the above, it's important to bear in mind that the interpreted topographies reflect difference waveforms - and thus, any dipole inferences hinges on the topographies reflecting an underlying "difference/mismatch" response and not just differences in different co-occurring responses. Hence, to guide the reader in all of the above, it would be of great help if the topographies of the underlying responses to the match and mismatch conditions for the reported time window together with the rather clear P1-N1-P2 complex could be visualized either in the main paper or in the supplementary material for reference?

REFERENCES:

Sassenhagen J, Draschkow D (2019). Cluster‐based permutation tests of MEG/EEG data do not establish significance of effect latency or location. Psychophysiology. e13335. https://doi.org/10.1111/psyp.13335

Rereading this section of the discussion, we agree wholeheartedly with the reviewer that it lacks a degree of detail both in terms of our interpretation and the descriptions of the effects. On reflection, we wondered if the best fit for this paragraph might be later in the discussion, specifically pairing it with the discussion of N400/P300 elicitation in instances of musical mismatch (497-507) and the concluding remarks. We’ve also substantially tamed down a few claims, and tried to reframe this so that it is a little more in-keeping with the general conclusions. Unfortunately it is difficult to cite similar research on the PMN using concurrently presented stimuli or non-linguistic input, because to our knowledge this doesn’t really exist. Beyond this, substantial differences in reporting of timing and topography of the PMN across papers and paradigms (sometimes due to paradigm/manipulation, other times due to recording or preprocessing, and finally, on other occasions not clearly caused by anything at all) mean that we don’t feel particularly confident drawing strong conclusions about the right lateralization of the topography of the effect. Finally, we have included the topography maps for all conditions (not just difference waves) in the supplementary material.

---

## [Editor Report · Decision Letter 2]

27 Nov 2024

The Phonological Mapping Negativity (PMN) as a language-specific component: exploring responses to linguistic vs musical mismatch.

PONE-D-23-09694R2

Dear Dr. Lewendon,

We’re pleased to inform you that your manuscript has been judged scientifically suitable for publication and will be formally accepted for publication once it meets all outstanding technical requirements.

Kind regards,

Laura Morett

Academic Editor

PLOS ONE

Additional Editor Comments (optional):

Apologies for the delay. I thank the authors for their attention to the remaining points raised by R2. I have reviewed their revisions and responses and have confirmed that they address the remaining concerns. Thus, the manuscript is now suitable for publication in PLOS One.
---

## [Editor Report · Acceptance letter]

2 Dec 2024

PONE-D-23-09694R2 

PLOS ONE

Dear Dr. Lewendon, 

I'm pleased to inform you that your manuscript has been deemed suitable for publication in PLOS ONE. Congratulations! Your manuscript is now being handed over to our production team.

Kind regards, 

on behalf of

Dr. Laura Morett 

Academic Editor

PLOS ONE